# Assessing the impact of timely diagnosis on psychological outcomes and quality of life for cancer patients: A scoping review

Laura Boswell[1], Jenny Harris[1], Richard Green[1‡], Jo Armes[1], Georgia B. Black[2*], Katriina L. Whitaker[1]

**1** School of Health Sciences, University of Surrey, Surrey, United Kingdom, **2** Wolfson Institute of Population Health, Queen Mary University of London, London, United Kingdom

☯ These authors contributed equally to this work.
‡ These authors also contributed equally to this work
* g.black@qmul.ac.uk

## Abstract

### Objective

We explored the literature on timely cancer diagnosis and its significance on psychological outcomes or quality of life in cancer patients.

### Design

A scoping review to map existing literature in this area, following the Preferred Reporting Items for Systematic Reviews and Meta-Analyses extension for Scoping Reviews.

### Results

Six studies were identified. Four studies used cross-sectional surveys, and one each used qualitative and mixed-method designs. Quantitative evidence suggests that timely diagnosis was associated with better psychological outcomes and quality of life. Qualitative and mixed-method evidence found an incidental relationship, but it was not a focus of the studies. There were varied definitions of timely diagnosis, and a diverse range of measures were used to identify outcomes. No study satisfied all quality appraisal criteria, with key dates in the diagnostic journey being the least reported (0/6 studies).

### Conclusions

Preliminary evidence indicates that timely diagnosis may be associated with variations in psychological outcomes and quality of life in patients with cancer; however, methodological heterogeneity restricts the generalisability of the findings. More

**Data availability statement:** All data underlying the findings are derived from publicly available, peer-reviewed articles referenced in the paper and accessible via academic databases.

**Funding:** GBB acknowledges funding from Barts Charity (G-001520) and The Institute of Healthcare Improvement Studies (University of Cambridge) (RG88620/PD-2019-02-004). The funders had no role in study design, data collection and analysis, decision to publish, or preparation of the manuscript.

**Competing interests:** The authors have declared that no competing interests exist.

high-quality longitudinal quantitative and qualitative research is needed to explore the direction of the association and lived experience during the adjustment process.

## Introduction

### Rationale

Timely diagnosis and treatment of cancer are considered crucial for improving outcomes and prognosis of patients with cancer [1]. These goals are recognised by the creation of policy guidance and targets which advocate for early detection at the national level [2–4]. Evidence shows that repeated help-seeking and protracted intervals can lead to emergency hospital presentation, resulting in poorer clinical and patient-reported outcomes, including diminished survival and worse quality of life (QoL), than those diagnosed through other routes, such as referral from primary care or participation in screening [5–7]. Currently, there is no clear definition of timely diagnosis in common use. Studies have often used the term to mean 'early-stage diagnosis' [8], however, a cancer is only deemed stageable if a system exists for a specific location where it originates in the body and the type of cells in the cancer. Alternative indicators relate to the promptness of diagnosis subsequent to the patient's clinical presentation or the healthcare services that the patient initially sought out [9].

The Model of Pathways to Treatment is a conceptual framework designed to structure and theorise studies relating to a timely cancer diagnosis [10]. This framework outlines disease-related, patient-related, and healthcare system-related factors in four distinct intervals: appraisal, beginning when an individual first becomes aware of bodily changes; help-seeking, from the decision that they need to discuss the symptom with a healthcare professional (HCP); diagnostic, from the first consultation to the diagnosis; and the pre-treatment interval, from diagnosis to the start of treatment. The framework highlights that a patient's journey through the pathway is not necessarily linear. For instance, individuals may re-enter the pathway after initially receiving an all-clear due to persistent or worsening symptoms. The model offers a valuable theoretical perspective for research, aiming to enhance the measurement, description, and comprehension of the time taken to diagnose and initiate treatment [11–13]. Expedited diagnostic pathways can be more challenging for some cancers with non-specific symptom signatures [14] or where the diagnostic process is particularly complex, such as when necessitating biopsy under general anaesthesia or accurate staging via multiple imaging modalities [15]. A systematic review of 209 studies by Neal et al. [16] examined the time to diagnosis concerning clinical and psychological outcomes, including survival, QoL, and patient satisfaction. Overall, they concluded that prolonged diagnosis had several negative consequences, such as later-stage diagnosis, poorer quality of life, and worse survival, although this was not universally found for all cancer types. They also concluded that efforts to improve the speed of diagnosis would lead to better patient outcomes. However, their study included only two studies with QoL as an outcome and did not consider wider psychological outcomes.

The COVID-19 pandemic exacerbated difficulties in timely diagnosis. Research has shown that this had a significant impact on symptomatic cancer diagnoses in terms of case identification, referral processes, and screening due to national programs being suspended, a route which accounts for approximately 5% of all cancer diagnoses [17]. One time-controlled study suggested that people diagnosed with cancer in a six-month period during COVID-19 were more likely to be diagnosed at a later stage than those diagnosed before the pandemic [18]. Despite experiencing symptoms that could indicate cancer, many individuals chose not to seek healthcare services during periods of lockdown and restricted healthcare access, as people reported being anxious about catching COVID-19 and overburdening the NHS for what they interpreted as trivial concerns [19]. This reasoning was partly caused by new regulations on referring patients to secondary care services and asking them to present with only major or urgent concerns [20]. However, hospital referrals and emergency presentations are associated with advanced stages at the time of diagnosis, as opposed to diagnoses made through routine primary and secondary care referral pathways and screening programs [21]. Prolonged diagnosis and treatment are expected to decrease the number of cancer cases in the short term, followed by an increase in advanced-stage diagnoses and mortality [22].

Cancer is a traumatic event which can significantly impact an individual's psychological well-being [23–25]. Much is known about the impact of various delays on survival; however, markedly less is known about the other outcomes that affect cancer survivors' experience, such as the emotional and psychological consequences for those living with cancer and beyond. These outcomes are important to understand, given that the number of patients who survive cancer is expected to rise due to the increased uptake of screening opportunities and improvements in treatments [26,27]. In addition to cancer being a potentially life-threatening illness, it is characterised by a potentially all-embracing impact on the psychological adjustment of survivors and subsequent presentations of psychological distress [28]. Despite improved survival rates, cancer survivors may experience psychological issues resulting from the disease and its treatments [29–31], which can have a profound impact on their perspective, self-image, and well-being [32]. Many studies have analysed depression and anxiety as primary psychological responses to cancer [33–37], however, the impact of timely cancer diagnosis on psychological outcomes remains unclear and under-researched.

Timely diagnosis has been suggested to have a positive psychological impact; research by Murchie et al. [38] found that although there was no significant relationship between provider delay and survival, timeliness matters as delays cause heightened anxiety and worry. Additionally, analysis of a free text item from the English Cancer Patient Experience Survey (n = 1056), where people were asked to provide any additional comments about any aspect of their experience, found that some people (n = 27) spontaneously mentioned that timely diagnosis or prolonged diagnosis was an important aspect of the patient experience [39]. A recent cross-sectional secondary analysis examined the relationship between diagnostic time intervals and health-related quality of life (HRQoL), anxiety, and depression in adolescents diagnosed with cancer [40]. This study suggests that those who experienced a prolonged diagnosis of cancer (defined by the time of the diagnostic interval) were more likely to have poorer health-related quality of life and exhibit symptoms of clinical anxiety or depression than those who experienced shorter diagnostic intervals. However, the extent to which these results can be translated to the adult population in terms of psychological outcomes and quality of life (QoL) is unclear.

## Objectives

This scoping review aimed to explore the literature related to timely cancer diagnosis and its significance on psychological outcomes or quality of life (QoL) in cancer patients.

The following research questions were identified:

• What types of psychological and QoL outcomes have been studied?

• Which assessment methods have been used to identify these outcomes?

- How have studies conceptualised timely diagnosis?
- What is the evidence of the relationship between timely diagnosis and psychological outcomes and/or QoL of cancer patients?

## Methods

A scoping review was chosen because it allowed us to map existing research conducted in this area [41]. Using this method, we provide a narrative synthesis of key ideas and concepts and identify gaps in the literature.

### Protocol

This review was created following the Preferred Reporting Items for Systematic Reviews and Meta-Analyses (PRISMA) extension for Scoping Reviews: Checklist and Explanation [42], and followed Arksey and O'Malley's methodological framework for scoping studies [41].

### Information sources and searching

Three electronic databases were searched: MEDLINE, Cumulative Index for Nursing and Allied Health Professionals (CINAHL), and PsycInfo, as these were pragmatically deemed to be the most relevant databases for the topic. Search terms were developed in consultation with a research librarian to include terms relevant to cancer, timely diagnosis, and psychological outcomes/QoL (S1 Table). All three databases were searched via EBSCOhost. MEDLINE and CINAHL were searched together, followed by a second search using PsycInfo. The most recent search commenced on the 25th of November 2024.

### Eligibility criteria

Any empirical studies that explored, measured, or focused on the time to cancer diagnosis, psychological outcomes, and the links between them were included in this review. Eligible studies were original research in English and published in peer-reviewed journals from January 2007 to November 2024. 2007 was chosen based on NHS England's publication of the Cancer Reform Strategy, which aimed to improve cancer services, including earlier diagnoses [2]. Included studies comprising adult cancer samples focused on the diagnostic process or perceived process, rather than the diagnostic outcome, such as the length of the diagnostic interval, rather than the stage of cancer. Studies were excluded if they did not consider psychological outcomes/themes or QoL, did not include adult individuals with a history of cancer (<18 years), or were only focused on the cancer stage (S2 Table).

### Selection of sources of evidence

Titles and abstracts were initially screened, and full-text articles were obtained and assessed (LB). In both rounds, 10% of the articles were screened by a coauthor (RG) to check reliability. Any uncertainty was resolved through discussion with two coauthors (JH and KW). Screening was completed using Rayyan software [43]. Systematic review reference lists were manually searched to identify additional original studies. One additional paper was identified through hand-searching.

### Data charting process

A standardised data-charting form was used to extract the data (LB), including the methodology, definitions, and measures used to identify gaps and shared themes. The resulting themes were discussed and reviewed by the full authorship team.

## Appraisal of sources of evidence

The authors conducted a quality appraisal, which mitigates the risk of bias, increases the relevance of the findings, and increases the likelihood of uptake into policy and practice [44]. We combined the Mixed Methods Appraisal Tool (MMAT) [45]; chosen as it was designed to assess all research methodologies, with the assessment criteria recommended by Neal et al.[16], whose systematic review identified high-quality standards for future research in this area. We used the Neal et al. assessment criteria for the current study, as this specifically relates to research examining patient perceptions of timeliness, psychological outcomes, and QoL. Neal et al.[16] identified studies capturing patient experience, QoL, and psychological outcomes, and recognised the need to increase knowledge in this area. They specified that 'good studies' will be reported, as described in S3 Table. The quality appraisal informed our critical evaluation of the existing evidence. An original rating was conducted by LB and then discussed through consensus and verified by two co-authors (JH and KW). All authors reviewed the appraisal tools to increase process accuracy.

## Results

### Identification of studies

In total, 4620 titles were screened for eligibility, of which 261 full texts were retrieved from databases, with one additional article found by searching reference lists (Fig 1). Six studies were included in the final review.

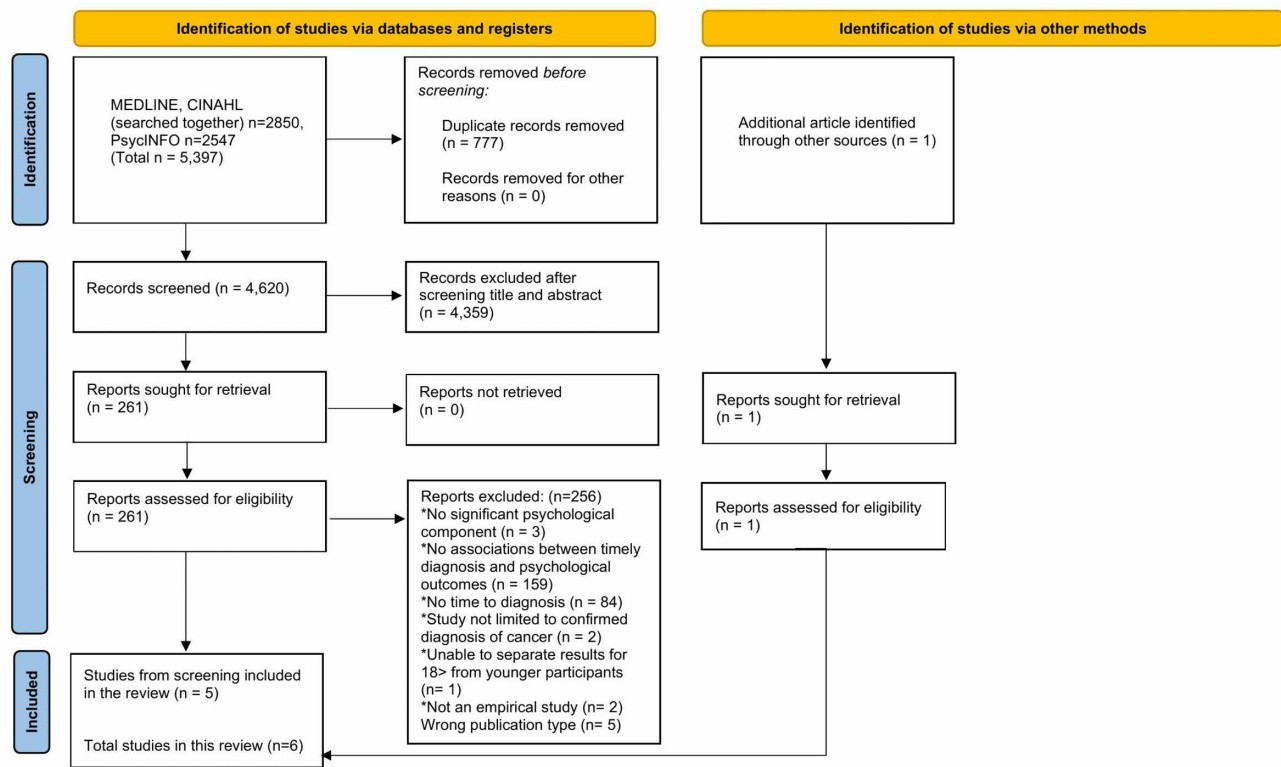

**Fig 1. PRISMA 2020 flow diagram for new systematic reviews, which included searches of databases, registers and other sources [42].**

## Study characteristics

Most included studies were cross-sectional survey designs (n = 4), with one study being qualitative and one mixed-method design (Table 1). All studies were published between 2011 and 2020. Two studies were conducted in the UK [46,47], one in Denmark [48], one in the Netherlands [49], one in the USA [50] and one was conducted in Sweden, Denmark, and England [51]. Across all studies, the minimum number of participants was 34, and the maximum number was 75,399.

The sex of the participants was reported in four out of six studies, with broadly equal representation of men and women overall. Colorectal cancer was the most included cancer type (n = 3) [46,47,51], with one study looking at colorectal cancer only [47], followed by lung cancer (n = 2) [50,51], with one study looking at this cancer only [50]. The other types included breast, prostate, endometrial, ovarian, and sarcoma [46,48,49]. Where reported, there was a wide range between studies in the timing of participants' diagnosis and the data collection ≤6 months to 12 years [47,48,51], with three studies not fully reporting this [46,49,50]. Four of the six papers specifically aimed to examine the association between timely diagnosis and psychological outcomes and/or QoL [46–49]. For two of the studies, both qualitative and mixed-methods, this was not a primary aim but an incidental finding [50,51].

## Quality appraisal

Quality appraisal was completed for all the included studies (S4 Table). All six studies met the MMAT criteria and yielded similar results across all studies. However, none of the studies met all the criteria established by Neal et al. [16] (Fig 2). In three of the six studies, time intervals were not reported [47,50,51]. The three studies that reported them did not provide dates [46,48,49]. One study used a proxy for timeliness: the number of consultations with a general practitioner (GP) prior to diagnosis (three or more), which adheres to the Aarhus statements' primary care interval [46]. Three studies (49–51) used retrospective recall as part of cross-sectional surveys. Two studies qualitatively reported the patients' narrative concerns [50,51]. However, as these studies were exploratory, they could not define the time intervals without previously identifying this concern.

Importantly, five studies gathered patient perceptions of timely diagnosis (or patient-reported intervals), in addition to having a previously defined definition [47–51]. In doing so, these studies gathered data recommended in the quality assessment by Neal et al. [16], which calls for future research to explore the connection between patients' perceptions of timeliness and their quality of life, in addition to psychological outcomes. Although none of the studies completely satisfied the quality criteria, the open-question nature of the studies examined the relationship between time intervals and psychological outcomes. None of the studies reported any dates related to events, such as bodily changes, initial presentations, diagnoses, or referrals.

## Psychological outcomes and QoL

There were four broad areas of different psychological outcomes and QoL indicators included in the studies, which across quantitative studies, were measured using six different scales, including QoL, fear of recurrence/distress, and coping. The other psychological constructs identified were more unstructured and presented as themes from qualitative and mixed-methods research.

**QoL.** QoL is a concept determined by needs, beliefs, values, and attitudes that contribute to a person's satisfaction and happiness [52]. Three studies reported QoL [47–49], making it the most common psychological outcome. The European Organisation for Research and Treatment of Cancer (EORTC) QLQ-C30 [53] was used twice. As a QoL instrument, the EORTC-QLQ-C30 comprises nine sub-scales: five pertaining to function (physical, role, cognitive, emotional, and social); three about symptoms (fatigue, pain, and nausea and vomiting); a global health and QoL life scale; and six single-item symptom measures. One study used this in a Danish translation [48]. The third study that measured QoL used the Functional Assessment of Cancer Therapy-Colorectal (FACT-C) [54]. This is a colorectal cancer-specific version of the FACT-G, a general QoL measure containing four well-being categories: physical, functional, social/family, and emotional.

 

**Table 1. Study characteristics.**

| 1st Author, Year of publication, and Country | Sample, Cancer type, Study design, and Analysis | Time since diagnosis | Definition of timely diagnosis | Was an association between timely diagnosis and psychological outcomes or QoL the primary objective? | Psychological concepts or outcomes measured/ explored | Psychological measure used or theme to emerge |
|---|---|---|---|---|---|---|
| Lehto, 2011, USA | N = 34, 35% female Lung Mixed methods – Cross-sectional | New diagnosis (no further details provided) | Patient quote – "Did it get diagnosed early enough"? | No | Described participants feeling uncertainty and being overwhelmed. They "felt lucky because of early diagnosis" | The conceptual content cognitive map (3CM)/ Cognitive Map Theory |
| Malmstrom, 2018, Sweeden, Denmark, and England | N = 155, 46% female Lung and Colorectal Qualitative – thematic analysis | ≤6-months diagnosis | Patient perceptions of timeliness, captured during qualitative interviews | Yes | Patients described fear and anxiety (participants who were left without information about how waiting times might affect their prognosis spoke of the fear and anxiety this provoked). Others said they felt "lucky […] I was dealt with quickly" | No measure was used for this analysis type |
| Mendonca, 2015, UK | N = 73,462, gender anonymised 24 cancer types. The three most common were breast, colon, and prostate Quantitative – Cross-sectional | Time not reported | No more than two consultations with a general practitioner (GP) | Yes | Patients who experienced a longer diagnostic interval reported a less positive experience of cancer care | The Cancer Patient Experience Survey |
| Miles, 2016, UK | N = 296, 51% female Colorectal Quantitative – Cross-sectional | 6 months to 12 years post diagnosis | 'Perceived diagnostic delay' - 'Do you think your cancer could have been diagnosed sooner than it was' with response options: 'yes', 'not sure' and 'no' | Yes | 'Cancer-related distress' – concern that any delay in diagnosis has resulted in more advanced disease necessitating more toxic or invasive treatments. It increases fear of cancer recurrence and raises concerns that the cancer and its treatment have had an adverse effect on the individual's quality of life. | QoL measured using FACT-C // Post-traumatic stress disorder measured using IES-R |
| Robinson, 2011, Denmark | N = 353, 100% female Endometrial, ovarian Quantitative – Cross-sectional | Diagnosis 5–6 years prior to interview | Total delay (number of weeks between first cancer symptom and initiation of treatment) | Yes | QoL (linked with patient satisfaction) | QoL was assessed using the standard instrument EORTC-QLQ-C30 in the Danish-validated translation |

*(Continued)*

**Table 1.** (Continued)

| 1st Author, Year of publication, and Country | Sample, Cancer type, Study design, and Analysis | Time since diagnosis | Definition of timely diagnosis | Was an association between timely diagnosis and psychological outcomes or QoL the primary objective? | Psychological concepts or outcomes measured/ explored | Psychological measure used or theme to emerge |
|---|---|---|---|---|---|---|
| Soomers, 2020, Netherlands | N = 1,099, 46% female Sarcoma Quantitative – Cross-sectional | Time not reported | 'Patient interval' - the time from the first symptom until the first presentation to a doctor and 'Diagnostic interval' - the time from this first presentation until pathologic diagnosis // This was assessed by a single question— "Do you think your HRQoL was influenced by your diagnostic interval length?" | Yes | HRQoL //Perceived Impact of Diagnostic Interval Length on HRQoL // Coping | Health-Related Quality of Life: EORTC QLQ-C30// Perceived Impact of Diagnostic Interval Length on HRQoL - This was assessed by a single question; "Do you think your HRQoL was influenced by your diagnostic interval length?" // Coping: Illness Cognition Questionnaire for chronic diseases (ICQ) |

| Appraisal Criteria | Lehto et al., 2011 | Malmstrom et al., 2018 | Mendonca et al., 2015 | Miles et al., 2016 | Robinson et al., 2011 | Soomers et al., 2020 |
|---|---|---|---|---|---|---|
| Does the study report time intervals in compliance with the definitions in the Aarhus statement? (52) | ✘ | ✘ | ✔ | ✘ | ✔ | ✔ |
| Does the study report dates (bodily change/presentation/diagnosis/referral)? | ✘ | ✘ | ✘ | ✘ | ✘ | ✘ |
| Did the study use appropriate data collection methods for the time intervals? | ✘ | ✘ | ✔ | ✘ | ✔ | ✔ |

**Fig 2. Studies in compliance with the Neal et al. appraisal criteria [16].**

**Fear of recurrence and distress.** The next most assessed concepts were fear of recurrence, luck, and cancer-related distress [47,50,51], which were measured using the Conceptual Content Cognitive Map (3CM) [55] and the Revised Impact of Events Scale (IESR) [56]. The 3 CM allows participants to note important concepts related to a specific issue (cancer perception and concerns before and after surgery, in this case), but then organise the concepts in a manner that is significant to the individual, thus creating a map. Miles et al. [47] chose to use the IESR specifically because it is one of the most commonly used measures of cancer-related distress. Cancer-related distress and suspected post-traumatic stress disorder (PTSD) were measured using the Revised Impact of Events Scale (IESR) [56].

**Coping.** Coping was another outcome measured using the Illness Cognition Questionnaire for Chronic Diseases (ICQ) [57]. People employ coping strategies such as acceptance, hope, and social support to adapt to situations resulting from stressful circumstances, such as a cancer diagnosis (67). Similar to the above measures, the ICQ is a multi-item (18 items) questionnaire comprising three main scales: helplessness as a way of emphasising the aversive meaning of the

disease, acceptance as a way of diminishing the aversive meaning, and perceived benefits as a way of adding a positive meaning to the disease (60). Other reported outcomes included uncertainty and feeling overwhelmed [50] measured again by the 3 CM [55]. A 'less positive experience' [46] was measured by the National Cancer Patient Experience Survey [58,59]. This concept is strongly tied to patients' expectations and whether they are positively realised.

**Other psychosocial concepts.** The National Cancer Patient Experience Survey offers individuals with cancer a means of providing feedback on various aspects of care. This evaluation is then utilised to enhance services at both the local and national levels, aiming to address individuals' requirements and preferences, and to implement enhancements based on what truly matters to the public. Critically to Mendonca et al. [46], who looked at overall satisfaction with care, the survey also reported the number of pre-referral General Practitioner consultations, which were used as a proxy for time to diagnosis.

Two studies employed qualitative data-collection methods. Lehto et al. [50] used mixed methods; however, the qualitative component involved an interview-like procedure using the 3 CM (55). In this process, participants are asked to contemplate and record the key ideas associated with a specific topic and then arrange them in a personally meaningful manner. Finally, Malmstrom et al. [51] used interviews for data collection and organised them thematically, counting anxiety as a psychological outcome; "*what has worried me is that everything took such a long time […] during the investigations… it always took several weeks in between [investigations]. So you thought that they had forgotten you, but they did not. They did not forget. But it was just the feeling that you did not hear anything*".

### Conceptualisation of timely diagnosis

The six studies conceptualised timely diagnosis in three different ways, including: time intervals, patient-reported definitions, and the number of pre-referral consultations as a quantitative proxy for timely diagnosis (S6 Table). Two quantitative studies reported time intervals; the number of weeks between the first cancer symptoms and the initiation of treatment [48], and found prolonged timeliness in the patient and diagnostic intervals [49]. Three studies used patient-reported definitions. One of these was quantitative and analysed time to diagnosis directly; for example, using the survey question, '*Do you think your cancer could have been diagnosed sooner than it was'?* [47] The other studies were qualitative and mixed methods in design, and both found the information incidentally.

The qualitative study asked, '*Could you start by telling me, in your own words and in as much detail as you want, about everything that has happened since you first started to suspect there might be a problem with your health?'* and was followed by probing questions [51]. Participants used temporal narratives to describe their experience of diagnosis (e.g., "better to be fast than slow", "it went like clockwork"), and this was interpreted as a sign of good care. Periods of time between investigations or care could be experienced as a "delay" in the absence of information that helped to contextualise the wait. The mixed methods study captured a patient quote 'Did it get diagnosed early enough'? [50]. Finally, one study used a quantitative definition, which was 'three or more pre-referral consultations', indicative of prolonged help-seeking/ diagnostic intervals [46].

### The influence of timely diagnosis on psychological outcomes and QoL

Across all studies in this review, there was evidence to suggest that receiving a timely diagnosis of cancer positively affects patients' psychological outcomes or QoL. In the two studies with a qualitative element [50,51], patients expressed feeling lucky that they received an earlier diagnosis or that they were dealt with quickly, "*and I suppose I maybe am one of the lucky ones, that I was dealt with quickly*" [51]. Both studies concluded that cancer-related distress was higher among participants who perceived a prolonged diagnosis, such as believing that there had been a medical error. Lehto et al. [50] described how participants had increased worry from questioning whether they had received a diagnosis 'early enough'. Malstrom et al. [51] mirror this, with their participants also expressing feeling lucky that they were dealt with quickly, yet

expand on this, demonstrating that patients often create temporal narratives that contextualise their feelings. Examples included *'better to be fast than slow'* and *'it went like clockwork'*.

In the quantitative research, one study concluded that the HRQoL of sarcoma survivors was correlated with the perceived impact of the diagnostic interval duration ($\beta = 0.087$, $p = 0.01$) but not with the actual length [49]. Similarly, another study found that longer total delays were also statistically significantly linked to worse patient satisfaction in the perceived diagnostic phase (coefficient not reported, $p < 0.01$), indicating that overall delays played a significant role in how patients felt about the care they received [48]. A potential lack of information could result in any period between examinations being perceived as a delay. The same study showed that QoL was significantly reduced by prolonged diagnosis (coefficient not reported, $p = 0.04$) in both ovarian and endometrial cancer patients [48].

One study reported that cancer-related distress (fear of cancer recurrence and QoL) was higher in those who perceived diagnostic delay [47]. Mendonca et al. [46] found that patients who had three or more consultations with a general practitioner reported having a more negative experience of cancer care (adjusted odds ratios ranged from 1.10–1.68, $p < 0.01$ for all) than those who had two consultations or fewer. However, Soomers et al. [49] found that more than half of their participants (58%) did not believe that the diagnostic interval length influenced their HRQoL. Patients who felt that their HRQoL was negatively affected (31%) had lower QoL scores than those who perceived a positive or no impact. Lower QoL scores were also associated with maladaptive coping strategies post-diagnosis, as indicated by scoring lower on acceptance and higher on helplessness.

Finally, Miles et al. [47] found that cancer survivors who perceived a prolonged diagnosis had higher levels of distress related to their cancer (adjusted $\beta$ 0.107, $p < 0.05$), while the disease stage at diagnosis and treatment did not have a significant impact on distress levels (four other studies also did not find an impact of cancer stage, on psychological outcomes) [48–51]. This suggests that the perception of a prolonged diagnosis, rather than the stage of diagnosis per se, is linked to adverse psychological outcomes.

## Discussion

To our knowledge, this is the first scoping review to systematically explore the literature on the association between timely diagnosis, psychological outcomes, and QoL in patients with cancer. Here, defined by the diagnostic interval, patient accounts, and number of pre-referral GP consultations, the findings indicate some preliminary evidence that timely diagnosis is associated with improved QoL and psychological outcomes, including fear, anxiety, and cancer-related distress.

We analysed six studies containing over 25 types of cancer, with colorectal cancer being the most represented cancer type across the six studies. The sample showed nearly equal representation of the sex of the population, with 49% being male. The qualitative studies showed that patients deduce significant meaning from the passage of time, and the effect of receiving a timely diagnosis on psychological well-being was positive. Temporal narratives are used to denote luck when receiving a timely diagnosis. Likewise, in the quantitative studies, QoL and patient satisfaction were reduced in patients who received diagnoses over extended intervals. Patients with three or more consultations reported more negative experiences across multiple domains of cancer care (for example, 'Care from your General Practice' and 'Your overall NHS care') when this was used as a proxy for a longer time to diagnosis [46]. These findings are congruent with those of previous studies, adding weight to Murchie et al. [38], who concluded that elevated levels of anxiety followed a deferral in diagnostic and treatment intervals. Likewise, the work concurs with the systematic review by Neal et al. [16], who observed that a timely diagnosis could lead to improved patient outcomes.

A novel finding of this review is the importance of how patients view the time taken to reach a diagnosis. In two of the six papers, patients' perception of the length of their diagnostic journey impacted their psychological outcomes and quality of life. One study reported that neither the stage of colorectal cancer at the time of diagnosis nor treatment type (surgery, radiotherapy, and chemotherapy) affected the level of distress, implying that it was the individual's perception of delay that was linked to negative psychological consequences [47]. Similarly, Soomers et al. [49] compared the association between

actual time to diagnosis and HRQoL and the perceived impact of the diagnostic interval length on HRQoL and found that participants who perceived a positive or no impact of diagnostic interval length had higher QoL than those who perceived a negative impact. They concluded that diagnostic interval length was not associated with HRQoL scores, but that the perceived impact of diagnostic interval length on HRQoL was.

Despite some evidence of a positive association between timely diagnosis and psychological outcomes, we identified several issues that limit the ability to draw robust conclusions. In this review, one study used a proxy for the timeliness of diagnosis [46], and five of the six studies obtained data from patient self-reports, for example, 'Do you think your cancer could have been diagnosed sooner than it was'? [47]. This introduces the potential for bias, particularly recall bias [60], especially where the participants had a long retrospective recall period of up to 12 years since their diagnosis [47]. This variation introduces further uncertainty, as participants' recall and interpretation of their diagnostic experiences and associated psychological impacts are likely to shift over time, affecting the reliability and comparability of the findings. None of the studies reported key dates along the diagnostic journey, as recommended by Neal et al. [16], which would have helped strengthen interpretations in either direction. Further, the studies varied in how they defined or measured 'timely diagnosis', using diagnostic intervals, number of pre-referral consultations, or patients' subjective perceptions of delay. This heterogeneity complicates the synthesis and may conflate objective timing with patients' emotional experiences. As such, subjective measures may reflect distress rather than actual diagnostic delay, potentially confounding associations with psychological outcomes. Conversely, interval-based approaches, while more structured, are limited by uncertainty around key time points like symptom onset and may not indicate whether an earlier diagnosis was realistically possible.

While intervals were the most common framing of timeliness, their main value lies in offering a broad structure for understanding the diagnostic journey between cancers and patient groups. In both qualitative and quantitative studies, identifying key milestones, such as first symptom recognition or first healthcare encounter, is often shaped by patients' memory, interpretation, and social contexts. Although intervals can suggest opportunities for earlier diagnosis, they cannot always capture the emotional impact or provide definitive evidence that a diagnosis could have occurred sooner. Given these challenges, future research would benefit from adopting consistent definitions of timely diagnosis, for example, by using the recommendations from Neal et al. [16], alongside greater attention on how patients interpret and emotionally respond to their diagnostic journey. This may involve capturing different aspects of timely diagnosis (e.g., using medical records or defined time intervals) as well as understanding patient experience (e.g., perceptions of the diagnostic process), thereby providing a more comprehensive view of both clinical performance and patient-centred outcomes.

As stated in the eligibility criteria, we focused our attention on the diagnostic process or perceived process rather than the diagnostic outcome, such as cancer stage. However, identifying a timely diagnosis using time intervals, especially for those who present as emergencies, is also precarious because of confounding principles such as the 'waiting-time paradox' [61]. Patients with late-stage disease are prioritised for quicker referral and treatment within a particular healthcare system than those with early-stage disease [62]. Further studies have concluded that short intervals cannot be strictly indicative of a timely diagnosis, resulting in increased survival [27–29]. As previously observed, it is well established that a cancer diagnosis with minimal moratorium is associated with higher survival rates. Building on this, we have shown the favourable effects of timely diagnosis on patients' psychological well-being and QoL.

Our findings indicate that patients' perception of diagnostic length results in worse psychological outcomes, which has implications for how healthcare organisations communicate and provide information about the medical process and the length of time they expect patients' medical journey to take. Again, this finding is consistent with previous research suggesting that more support be provided to individuals who have been diagnosed with severe and life-threatening illnesses, with a particular focus on helping them cope with the emotional fallout that may follow [39]. Clinicians and policymakers can use this information to assess current practices in patient communication and the expectation management of likely timeframes.

## Strengths and limitations

This review followed Arksey and O'Malley's methodological framework for scoping reviews [41]. The use of a predetermined structure ensured that the review was rigorous and reproducible. Another strength is the amalgamation of validated and recommended criteria to create a tailored quality assessment. We augmented the mixed-method critical appraisal tool [45] with a bespoke approach specific to psychological outcomes [16]. Doing this ensured that we were able to accomplish the most suitable appraisal of the current review under our objective. The limitations of the final study include the possibility of missing studies in foreign language materials and those that focused on the psychological outcomes of timely diagnosis in children and adolescents under 18 years of age, and those that may have been identified through searching via more databases. Only one of the included papers mentioned diagnosis resulting from screening procedures; however, this diagnostic method was not discussed further in terms of psychological outcomes. Thematic synthesis was not possible because of the diverse nature of the studies.

Several research gaps exist in the literature. Only the qualitative studies we included were specifically designed to examine diagnostic intervals and psychological outcomes. Therefore, all qualitative and mixed methods literature only identified this outcome incidentally. Additionally, none of the included studies reported key dates in the diagnostic journey, which would strengthen evidence of the relationship between time to diagnosis and psychological outcomes. The observational design of the included studies linking psychological outcomes and QoL to timely cancer diagnosis did so without examining the psychological processes that led to them. Other factors may also play a role, such as socio-demographic characteristics, cancer type, treatment type, or healthcare system factors, and these have not yet been studied in the context of understanding the impact of early diagnosis on psychological outcomes and QoL for cancer patients. The cancer stage at diagnosis was excluded from our review as we were interested in the process of diagnosis. However, it is recognised that this could also affect psychological outcomes.

Because of the recall bias that may have occurred in some of the studies, solutions to mitigate this in future work include reducing the length of the recall period [63] and conducting test-retest assessments [64]. Alternatively, longitudinal studies could utilise methods such as reflective journals, allowing patients to record data in real time, although this is not appropriate (or possible) for every study. This methodology would also support longitudinal studies that track changes over time to discover the processes of adjustment to a timely cancer diagnosis through post-treatment survivorship. It was beyond the scope of this review to differentiate between long-term and short-term psychological outcomes, and this is recommended for future research. This scoping review has revealed a need for more high-quality quantitative longitudinal studies and standardised measures in line with the quality criteria outlined by Neal et al. [16]. Additionally, to determine what to measure and in what ways, a critical first step would be to lay a qualitative groundwork to understand the nature and nuances of the experience of the timeliness of diagnosis.

## Conclusion

Preliminary evidence indicates that timely diagnosis may be associated with variations in psychological outcomes and QoL in patients with cancer; however, there is much uncertainty in the evidence and not yet enough of it. Our findings indicate that QoL and other outcomes may be affected, yet with much methodological and clinical heterogeneity in the study design, it is difficult to draw a strong conclusion. The current findings lack standardised qualitative and quantitative research with consistent reporting methods and definitions to explore the direction of the association and lived experience in the adjustment process. Further research focusing on patients' psychological adjustment is required to understand this process over time.

## Supporting information

**S1 Table. Search strategy.**
(DOCX)

**S2 Table. Inclusion and exclusion criteria.**
(DOCX)

**S3 Table. Neal.**
(DOCX)

**S4 Table. Qual. App**
(DOCX)

**S5 Table. PRISMA checklist.**
(DOCX)

**S6 Table. Conceptualisation of timely diagnosis.**
(DOCX)

## Author contributions

**Conceptualization:** Laura Boswell, Jenny Harris, Jo Armes, Georgia B Black, Katriina L. Whitaker.

**Data curation:** Laura Boswell.

**Formal analysis:** Laura Boswell, Jenny Harris, Richard Green, Georgia B Black, Katriina L. Whitaker.

**Funding acquisition:** Jenny Harris, Katriina L. Whitaker.

**Investigation:** Laura Boswell.

**Methodology:** Laura Boswell.

**Supervision:** Jenny Harris, Richard Green, Jo Armes, Georgia B Black, Katriina L. Whitaker.

**Writing – original draft:** Laura Boswell.

**Writing – review & editing:** Laura Boswell, Jenny Harris, Richard Green, Jo Armes, Georgia B Black, Katriina L. Whitaker.

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
