## [Decision Letter · Decision Letter 0]

20 May 2025

Assessing the impact of timely diagnosis on psychological outcomes and quality of life for cancer patients: a scoping review

PLOS ONE

Dear Dr. Black,

Thank you for submitting your manuscript to PLOS ONE. After careful consideration, we feel that it has merit but does not fully meet PLOS ONE’s publication criteria as it currently stands. Therefore, we invite you to submit a revised version of the manuscript that addresses the points raised during the review process.

We look forward to receiving your revised manuscript.

Kind regards,

Prof. Ritesh G. Menezes, M.B.B.S., M.D., Diplomate N.B.

Academic Editor

PLOS ONE

Journal Requirements:

[GBB acknowledges funding from Barts Charity (G-001520) and The Institute of Healthcare Improvement Studies (University of Cambridge) (RG88620/PD-2019-02-004).].

4. Please note that your Data Availability Statement is currently missing the repository name and/or the DOI/accession number of each dataset OR a direct link to access each database. If your manuscript is accepted for publication, you will be asked to provide these details on a very short timeline. We therefore suggest that you provide this information now, though we will not hold up the peer review process if you are unable.

5. Please include captions for your Supporting Information files at the end of your manuscript, and update any in-text citations to match accordingly. Please see our Supporting Information guidelines for more information: http://journals.plos.org/plosone/s/supporting-information .

Reviewers' comments:

Reviewer's Responses to Questions

**Comments to the Author**

1. Is the manuscript technically sound, and do the data support the conclusions?

Reviewer #1: Partly

Reviewer #2: Yes

2. Has the statistical analysis been performed appropriately and rigorously?

Reviewer #1: N/A

Reviewer #2: N/A

3. Have the authors made all data underlying the findings in their manuscript fully available?

Reviewer #1: Yes

Reviewer #2: Yes

4. Is the manuscript presented in an intelligible fashion and written in standard English?

Reviewer #1: Yes

Reviewer #2: Yes

Reviewer #1: This scoping review addresses an important and under-researched topic: the relationship between timely cancer diagnosis and psychological outcomes or quality of life (QoL) in adult cancer patients. The manuscript is well-structured, adheres to the PRISMA-ScR guidelines, and provides a clear narrative synthesis of the limited evidence available. The authors successfully map the existing literature, identify methodological gaps, and highlight the need for further research. However, there are areas where the manuscript could be strengthened to enhance clarity, rigor, and impact. Below, I provide detailed feedback for revisions.

1. The manuscript notes significant variability in how "timely diagnosis" is defined across studies (e.g., diagnostic intervals, number of pre-referral consultations, patient perceptions). While this heterogeneity is acknowledged, the discussion could be strengthened by explicitly addressing how these differing definitions impact the interpretation of findings. For example, patient-reported perceptions of delay may reflect emotional distress rather than objective time intervals, potentially confounding the association with psychological outcomes.

2. The quality appraisal using the MMAT and Neal et al. criteria is a strength, but the manuscript notes that no study met all Neal et al. criteria, particularly regarding the reporting of event dates (0/6 studies). This is a significant limitation, as precise temporal data are critical for assessing timeliness. The manuscript does not sufficiently explore how this gap affects the reliability of the findings or the feasibility of implementing these criteria in practice.

3. The manuscript briefly mentions socio-demographic characteristics as potential unstudied factors but does not discuss other critical confounders, such as cancer type, stage, treatment intensity, or healthcare system factors (e.g., access to care, referral pathways). These factors could significantly influence both the timeliness of diagnosis and psychological outcomes/QoL.

4. The review identifies a range of psychological outcomes (e.g., QoL, fear of recurrence, distress, coping), but the synthesis does not sufficiently differentiate between short-term (e.g., anxiety during diagnosis) and long-term (e.g., post-treatment distress) outcomes. This distinction is important for public health interventions, as the timing and nature of support may vary.

5. Add a brief analysis in the Results or Discussion that categorizes outcomes by temporal context (e.g., peri-diagnostic vs. survivorship) and discusses implications for intervention design. If the included studies do not provide enough data for this, acknowledge this as a gap.

6. Revise the abstract to briefly clarify the primary sources of heterogeneity (e.g., "varied definitions of timely diagnosis and diverse outcome measures").

7. Provide a complete financial disclosure statement, including author initials, grant numbers, funder names, URLs, and a statement on the funder’s role in the study.

8. "Data Availability Statement" Revise the statement to: "All data underlying the findings are derived from publicly available, peer-reviewed articles referenced in the manuscript and accessible via academic databases."

9. Ensure all references follow PLOS ONE’s style guide, including complete details for web-based sources (e.g., full URL, access date).

10. The manuscript makes a valuable contribution to the literature on timely cancer diagnosis and its psychological impacts. Addressing the major revisions (clarifying timely diagnosis definitions, deepening the quality appraisal discussion, exploring confounders, and differentiating outcome temporality) and minor revisions (improving abstract clarity, and submission requirements) will significantly enhance its rigor and readability. I commend the authors for their thorough approach and look forward to seeing the revised version.

Reviewer #2: The authors have researched a relevant research question exploring the timely diagnosis, psychological outcomes, and QoL in patients with cancer.

More details in the methods section would add value to the manuscript.

Overall English copyediting to avoid syntax errors is recommended.

Following are the detailed comments.

Line 38: dates of events being the least reported – please clarify. The phrase is not clearly understood.

Line 62: than those diagnosed through other routes – please specify the other routes

Lines 107-108: Much is known about the impact of various delays on survival; however, markedly less is known about the other outcomes that affect cancer survivors – please define other outcomes

Line 156: why were Scopus, Web Of Science & Embase databases not included for Literature search?

Line 161: Eligibility criteria – please specify the types of studies included as per study design, time duration (2007-2024?) for which studies were included, geographical location or any other specific inclusion & exclusion criteria that were employed

Line 169-170: did not include adult cancer samples (<18 years) – do you mean adult individuals? – please rephrase for better clarity

Line 187: designed to assess mixed- method research – please provide details/rationale for using mixed methods appraisal tool

How was the quality of studies assessed?

Lines 208-209: Most included studies were cross-sectional survey designs (n=4), with one study being qualitative and one mixed-method design – so did the review include only 6 eligible studies?

Lines 215-216: with one study looking at this cancer only – please specify the type of cancer

Table 1: study characteristics – 1st study, when n=34 – how is the design & analysis reported as case report? Please clarify

Similarly – quantitative case report for study by Robinson 2011 – please clarify the appropriateness of study design

Soomers 2020 – study cannot be both cross-sectional & cohort. Please check

Lines 315-316: Finally, one study used a quantitative definition of three or more pre-referral consultations. – provide details of the definition

Lines 366-367: We analysed six studies containing over 25 types of cancer, with colorectal cancer being the most common in three studies – try to avoid superfluous statements - 3 out 6 is only 50% & it would not be appropriate to say colorectal cancer to be most common.

Line 436: included papers mentioned screening detection – please clarify the meaning of the phrase

To check the uniform reporting & formatting of all references

**Do you want your identity to be public for this peer review?** For information about this choice, including consent withdrawal, please see our Privacy Policy

Reviewer #1: No

Reviewer #2: No

---

## [Decision Letter · Decision Letter 1]

20 Aug 2025

Dear Dr. Black,

Thank you for submitting your manuscript to PLOS ONE. After careful consideration, we feel that it has merit but does not fully meet PLOS ONE’s publication criteria as it currently stands. Therefore, we invite you to submit a revised version of the manuscript that addresses the points raised during the review process.

We look forward to receiving your revised manuscript.

Kind regards,

Prof. Ritesh G. Menezes, M.B.B.S., M.D., Diplomate N.B.

Academic Editor

PLOS ONE

Journal Requirements:

Academic Editor Comments:

• Methods: Mention the specific date when the last literature search was conducted.

• Methods: Description of the identification of records by snowballing from previous studies should be provided in the main text. This is illustrated in the PRISMA flowchart.

• PRISMA flowchart: Mention the specific number of records identified at MEDLINE, CINAHL, and PsycINFO (First box on the left-hand corner of the screen).

• PRISMA flowchart: Cross-check all the numbers provided in this Figure for accuracy.

• Supplementary File 1/Methods: Was the same string of search terms employed in all 3 databases? Related information must be provided in the main text. Besides, the information provided in the Supplementary File 1 lacks clarity.

• Page 20, Line 171, S2_In.Exclusion_Criteria: Indicate “Inclusion & Exclusion Criteria”. Similar corrections should be made elsewhere.

• Supplementary File 2: The presentation of the inclusion and exclusion criteria should be changed. Do not place them side by side in 2 columns. Place the list of exclusion criteria below the list of inclusion criteria, clearly differentiating between the two.

• Page 39: Provide the complete legend/caption/title for the Supplementary Files.

• Supplementary File 5: The footnotes provided should correspond to the information provided in the ‘Checklist’. For example, there seems to be no mention of JBI in the ‘Checklist’.

Reviewers' comments:

Reviewer's Responses to Questions

**Comments to the Author**

Reviewer #1: (No Response)

Reviewer #2: All comments have been addressed

2. Is the manuscript technically sound, and do the data support the conclusions?

Reviewer #1: (No Response)

Reviewer #2: Yes

3. Has the statistical analysis been performed appropriately and rigorously?

Reviewer #1: (No Response)

Reviewer #2: N/A

4. Have the authors made all data underlying the findings in their manuscript fully available?

Reviewer #1: (No Response)

Reviewer #2: Yes

5. Is the manuscript presented in an intelligible fashion and written in standard English?

Reviewer #1: (No Response)

Reviewer #2: Yes

Reviewer #1: As per the publication criteria of PLOS One, https://journals.plos.org/plosone/s/criteria-for-publication , the scoping review does not meet the criteria for publication.

Reviewer #2: The reviewer comments have been addressed.

Please Check the following references - reference numbers - 6, 7, 8, 9, 13, 14, 15, 18, 36, 39, 43, 45, 50, 62.

PMID not uniformly reported. Recommended to follow uniform referncing style.

**Do you want your identity to be public for this peer review?** For information about this choice, including consent withdrawal, please see our Privacy Policy

Reviewer #1: No

Reviewer #2: No

---

## [Author Response · Author response to Decision Letter 2]

29 Sep 2025

*Methods: Mention the specific date when the last literature search was conducted.

*Line 160 has now been updated to specify the last literature search date of the 25th of November.

*Methods: Description of the identification of records by snowballing from previous studies should be provided in the main text. This is illustrated in the PRISMA flowchart.

*Line 205 of the manuscript and the PRISMA flowchart have now been updated for clarity.

*PRISMA flowchart: Mention the specific number of records identified at MEDLINE, CINAHL, and PsycINFO (First box on the left-hand corner of the screen).

*The PRISMA flowchart has now been updated to illustrate that MEDLINE and CINAHL (searched together) retrieved 2850 articles, and PsycINFO retrieved 2547.

*PRISMA flowchart: Cross-check all the numbers provided in this Figure for accuracy.

*Thank you for highlighting this. The number of reports excluded has been updated.

*Supplementary File 1/Methods: Was the same string of search terms employed in all 3 databases? Related information must be provided in the main text. Besides, the information provided in the Supplementary File 1 lacks clarity.

*Thank you for this. Both the text from line 159 and the supplementary file have now been updated to clarify the process.

*Page 20, Line 171, S2_In.Exclusion_Criteria: Indicate “Inclusion & Exclusion Criteria”. Similar corrections should be made elsewhere.

*All instances of using this phrase (including the supplementary materials) have now been updated to ‘Inclusion & Exclusion Criteria’.

*Supplementary File 2: The presentation of the inclusion and exclusion criteria should be changed. Do not place them side by side in 2 columns. Place the list of exclusion criteria below the list of inclusion criteria, clearly differentiating between the two.

*This table has now been split into two separate tables, with the exclusion criteria below the inclusion criteria, as requested.

*Page 39: Provide the complete legend/caption/title for the Supplementary Files.

*All supplementary articles have been modified.

*Supplementary File 5: The footnotes provided should correspond to the information provided in the ‘Checklist’. For example, there seems to be no mention of JBI in the ‘Checklist’.

*This was an oversight and has been updated. Thank you for highlighting this.

*Please Check the following references - reference numbers - 6, 7, 8, 9, 13, 14, 15, 18, 36, 39, 43, 45, 50, 62. PMID not uniformly reported. Recommended to follow uniform referncing style.

*References should all now be correct and conform to PLOS One’s referencing style.

---

## [Editor Report · Decision Letter 2]

23 Oct 2025

Assessing the impact of timely diagnosis on psychological outcomes and quality of life for cancer patients: a scoping review

PLOS ONE

Dear Dr. Black,

Thank you for submitting your manuscript to PLOS ONE. After careful consideration, we feel that it has merit but does not fully meet PLOS ONE’s publication criteria as it currently stands. Therefore, we invite you to submit a revised version of the manuscript that addresses the points raised during the review process.

We look forward to receiving your revised manuscript.

Kind regards,

Prof. Ritesh G. Menezes, M.B.B.S., M.D., Diplomate N.B.

Academic Editor

PLOS ONE

Journal Requirements:

Additional Academic Editor Comments:

The minor comments are addressed by the authors. However, the major issue remains the lack or inconsistency of the definition or explanation for "timely diagnosis" of cancer. The related information put up in Table 1 is confusing and lacks clarity (specifically, the use of phrases like "time since diagnosis" and "ED" in the first row). Make sure to rearrange the order of columns. Besides, club information related to "authors, year of publication, country" in one column, and information on "sample, cancer type, study design, analysis" (follow the same sequence for all the references) in another column. The answers to the question "How have studies conceptualised timely diagnosis?" must be detailed in the text ('results' section). Provide more related information. Include a separate new Table summarizing the findings related to conceptualization of timely diagnosis in the studies that met the inclusion criteria. The related limitations must be further elaborated in the 'limitations' section. In relation to the definition/explanation of/for "timely diagnosis" of cancer, what are the recommendations suggested (by the authors of the present review) to be followed in future studies?

---

## [Author Response · Author response to Decision Letter 3]

4 Nov 2025

Thank you for your additional comments. We have now made the required amendments as advised, according to the guidance, and included a full description of responses in the attached files.

---

## [Editor Report · Decision Letter 3]

18 Nov 2025

Assessing the impact of timely diagnosis on psychological outcomes and quality of life for cancer patients: a scoping review

PONE-D-24-59414R3

Dear Dr. Black,

We’re pleased to inform you that your manuscript has been judged scientifically suitable for publication and will be formally accepted for publication once it meets all outstanding technical requirements.

Kind regards,

Prof. Ritesh G. Menezes, M.B.B.S., M.D., Diplomate N.B.

Academic Editor

PLOS ONE

---

## [Editor Report · Acceptance letter]

PONE-D-24-59414R3

PLOS One

Dear Dr. Black,

I'm pleased to inform you that your manuscript has been deemed suitable for publication in PLOS One. Congratulations! Your manuscript is now being handed over to our production team.

Kind regards,

on behalf of

Professor Ritesh G. Menezes

Academic Editor

PLOS One